



# On the role of building value models for flood risk analysis

Veronika Röthlisberger[1,2], Andreas P. Zischg[1,2], Margreth Keiler[1]

[1]Institute of Geography and Mobiliar Lab for Natural Risks, University of Bern, Hallerstrasse 12, CH-3012 Bern, Switzerland

[2]Oeschger Centre for Climate Change Research, University of Bern, Falkenplatz 16, CH-3012 Bern, Switzerland

*Correspondence to*: Veronika Röthlisberger (veronika.roethlisberger@giub.unibe.ch)

**Abstract.** Quantitative flood risk analyses support decisions in flood management policies that aim at cost efficiency. Risk is commonly calculated by a combination of the three quantified factors: hazard, exposure, and vulnerability. Our paper

focuses on the quantification of exposure, in particular on the relevance of building value estimation schemes within flood exposure analyses at regional to national scales. We compare five different models that estimate the values of flood exposed building. Four of them refer to individual buildings, whereas one is based on values per surface area, differentiated by land use category. That one follows an approach commonly used in flood risk analyses at regional or larger scales. Apart from the underlying concepts, the five models differ in complexity, in data and in computational expenses required for parameter

estimations, and in the data they require for model application.

The model parameters are estimated by using a database of more than half a million building insurance contracts in Switzerland, which are provided by 11 (out of 19) Cantonal insurance companies for buildings that operate under a monopoly within the respective Swiss Cantons. Comparing the five models' results with the spatially referenced insurance data applied directly suggests that models based on individual buildings produce better results than the model based on

surface area, but only when they include an individual building's volume.

Applying the five models to all of Switzerland produces results that are very similar with regard to the spatial distribution of exposed building values. Therefore, for spatial prioritizations, simpler models are preferable. In absolute values, however, the five models' results differ remarkably. The two simplest models underestimate the overall exposure, and even more so the extreme high values, upon which risk management strategies generally focus. In decision-making processes based on

cost-efficiency, this underestimation would result in suboptimal resource allocation for protection measures. Consequently, we propose that estimating exposed building values should be done based on individual buildings rather than on areas of land use types. In addition, a building's individual volume has to be taken into account in order to provide a reliable basis for cost-benefit analyses. The consideration of other building features further improves the value estimation. However, within the context of flood risk management, the optimal value estimation model depends on the specific questions to be answered.

The concepts of the presented building value models are generic. Thus, these models are transferrable with minimal adjustments according to the application's purpose and the data available.



## 1 Introduction

Flood damage accounts for a large proportion of the economic losses due to natural hazards in developed countries, e.g., for approximately one-third of the recent decades' losses in Switzerland (Bundesrat, 2016) and Europe (European Environment Agency, 2017). Flood losses are expected to increase not only due to ongoing anthropogenic climate change (IPCC 2014)

but also due to socioeconomic development (Kundzewicz et al., 2014; Liu et al., 2015; Arnell and Gosling., 2016). Future flood losses can be managed and ideally reduced with a wide range of measures. Yet, measures entail costs, either in the form of direct construction expenditures or, indirectly, through lost profits due to restricted land use. However, budgets are generally limited and thus they require measures be prioritized. This prioritization is based on quantitative flood risk analyses in many countries (European Parliament, 2007; Bründl et al., 2009). In this context, risk is commonly defined as a

combination of hazard, exposure, and vulnerability (see Birkmann, 2013 for an overview). It is usually expressed as the annual expected loss within a given area. Our paper focuses on exposure, in particular on the relevance of building value estimation schemes within flood exposure analyses at regional to national scales. However, as most risk analyses take the value of exposed assets into account in a linear way, this study's results have direct implications for flood risk analyses, too.

Different studies (e.g., de Moel and Aerts, 2011, Koivumäki et al., 2010) show that uncertainties in quantitative flood risk analyses are driven rather by uncertainties in the value of exposed assets than by uncertainties in area or frequency of floods. This is especially true at regional to national scales, where data availability limits the spatial resolution and differentiation of asset values within flood exposure analyses. Aggregated classes of land use have been the norm (Gerl et al., 2016), at least until recently, and the area specific value of each land use class is derived from lumped economic data of administrative

units (Merz et al., 2010). This transformation of values per administrative unit into values per spatial unit differentiated by land use class implies spatial data disaggregation, also referred to as dasymetric mapping (Chen et al., 2004; Thieken et al., 2006). While several case studies investigate the influence different data sources of asset values have on flood loss estimation (e.g., Bubeck et al., 2011; Budiyono et al. 2015, Cammerer et al., 2013; Jongman et al., 2012), the effect of dasymetric mapping methods is only addressed in a few publications. For instance, Wünsch et al. (2009) and Molinari and

Scorzini (2017) show in local case studies that even though the way in which exposed assets are estimated influences the resulting flood loss and thus flood risk, the spatial resolution of the exposed assets is more important. In both cases, the validation with recorded losses suggests that finer resolution of asset data improves the modelling results. Yet, both research teams conclude that further research on the impact of data resolution and disaggregation is needed. In fact, based on the growing availability of high resolution data and increasing computational power, more and more flood risk related studies at

national scales are based on data at the building level (e.g. Fuchs et al., 2015; Fuchs et al., 2017; Jongman et al., 2014, Röthlisberger et al., 2017). However, the individual monetary value of the buildings is usually not available due to data privacy restrictions and, thus, has to be estimated. There are different methods used in flood risk analyses to estimate individual building values. They range from uniform average value per building to sophisticated regression models




considering different building features. Yet, the role of these value estimation methods in flood risk assessments has received even less attention than the effect of dasymetric mapping methods. To the best of our knowledge, no study has compared different object-based building value models, nor have these object-based methods ever been contrasted with the commonly used approaches of land use specific values per area within the context of regional or national risk analyses. To fill this gap,

we investigate the influence of five different value estimation models (called M1 to M5) on the resulting values of flood-exposed buildings in Switzerland. Four of these models (M1, M2, M4 and M5, see upper most row in Table 1) refer to individual buildings, whereas one model (M3) uses average values of buildings per area, differentiated by land use category. The five models' underlying concepts are widespread in risk management, construction industry, and/or real estate management (see bottom row in Table 1), although they mainly differ in their complexity and requirements on data

resolution and differentiation.

However, this paper does more than evaluate building value models' role within flood risk analyses. Our study also investigates the models' influence on flood risk management decisions. In the context of the above mentioned need for prioritization, most current flood management policies aim at cost efficiency. With regard to cost-efficient measures, the actual monetary value of flood-exposed buildings is important, as are the statistical and spatial distribution of these values.

While the spatial distributions suggest areas of priority for the implementation of cost-efficient protection measures, the monetary values of exposed buildings affect the upper cost limits of such measures. Thus, we investigate both the monetary values and their distributions. As for distributions and actual values, the extremely high values are particularly relevant for risk management. Therefore, our study analyzes them in detail.

## 2 Methods applied and data used

The data and methods section is organized as follows. The first subsection (2.1) explains generically the set-ups of the five building values models and the estimation of their parameter values. In subsection 2.2, we describe subsequent steps towards values of flood exposed buildings, namely, the intersection with flood hazard maps and the results' spatial aggregation. The models are compared in subsection 2.3. The data used in this study are described in the last part of this section, subsection 2.4. Table 1 gives an overview of the five models with respect to their underlying concepts, data, and applications.

### 25 2.1 Models' set-up for value estimation

The five models in our study follow two different approaches. M3 is based on average value of buildings per area, differentiated by land use category. The other four models (M1, M2, M4, and M5) refer to individual buildings. These four models are defined as follows: M1, uniform average value per building; M2, uniform average value per building volume; M4, average value per building volume, differentiated by building features; and M5, value per building, individually

calculated based on linear regression. In the following, we outline the concepts of the five models and the estimation of their parameter values.



**Model M1: uniform average value per building**

Model M1 takes a straightforward approach as it assigns the same uniform average building value to each building. The parameter estimation requires two quantities with the same spatial aggregation, e.g. administrative units: (1) the total cumulative value, and (2) the total number of buildings within the same area. By dividing the total building value by the total

5   number of buildings, we obtain the value of the model's only parameter. The parameter corresponds to the average value of the buildings situated within the observed area. The unit of the M1 parameter is monetary value per building, e.g. [CHF].

**Model M2: uniform average value per building volume**

Model M2 is based on the building volumes only. The data requirements for the parameter estimation are similar to the ones of M1. In place of the total number of buildings, M2 requires the total cumulative volume of buildings within a given area.

10   To obtain the value of model's only parameter, the total building value is divided by the total building volume. Thus, the parameter of M2 is defined as the average value per building volume and is given in monetary value per unit volume, e.g. [CHF/m$^3$].

**Model M3: average building values per area, differentiated by land use category**

Model M3 takes a very common approach to flood risk analyses at national scales. It makes use of average building values

15   per unit area, differentiated by land use category. For the same given area, the parameter estimation requires two comprehensive data sets of comparable spatial resolution: (1) gapless polygons of land use types, and (2) spatially referenced data on building values. The two data sets are spatially joined, and the total building values per land use category are then calculated. In a last step, the cumulative building values per each type of land use are divided by the respective total area. This results in land use specific values of the model's parameter. They correspond to the average monetary building value

20   per area of each land use category, which is given in monetary value per unit area, e.g. [CHF/m$^2$].

**Model M4: average values per building volume, differentiated by land use category and building purpose**

Model M4's parameter is the same as in M2, i.e., the average monetary value per building volume. In contrast, the parameter values of M4 are not uniform but differentiated according to building features. In this study, land use category and building purpose are the criteria for differentiation. To estimate the specific parameter values of M4, we combine data on monetary

25   value, volume, land use category, and building purpose at the building level. These assignments at building level require inputting data of high spatial resolution and precise localization. For estimating M4's parameter values, the data assignments have to be complete for each individual building. However, in contrast to M1, M2 and M3, the input data for M4 do not need to be comprehensive within a given area. For M4, only buildings with complete information on value, volume and the differentiation criteria are considered, whereas the value and volume of all buildings from the same combination of

30   differentiation criteria (e.g. same land use category and building purpose) are summed up. Finally, the cumulated monetary



values are divided by the respective volumes, resulting in the model's parameter values. Thus, we obtain one specific value for each combination of differentiation criteria. The parameter's unit is monetary value per unit volume, e.g. [CHF/m$^3$].

**Model M5: value per building, individually calculated based on linear regression**

M5 is a linear regression model and is set up with the same input data as M4. We develop M5 in an exploratory manner by

starting with a maximal model, which includes all available explanatory variables, i.e. building features (Table 1 and Table A1), and their interactions. It is then reduced to simpler models by removing non-significant interactions and variables. In addition, models with transformed variables are set up. Out of this variety of models, we select the minimal adequate model. Namely, we follow the principle of parsimony and choose a model with a relatively small AIC (Akaike, 1974), a high coefficient of determination (adjusted R$^2$), and a minimal number of not-significant explanatory variables and interactions. In

addition, we plot the model's residuals to check visually if principal assumptions of linear regression on residuals are satisfied. The result of this exploratory process is the minimal adequate model that makes it possible to calculate the expected monetary value of a building as a linear function of the selected buildings attributes and interactions. This value is given in monetary units, e.g. [CHF].

While the five applied models are conceptually different, the estimation of their parameter values in our study is possible

based on the same data sets. Nevertheless, the parameter estimation is based on two different kinds of data subsets. This is because the first three models (M1 to M3) require a data selection, which fulfils different criteria in comparison to the selection for M4 and M5. While the crucial prerequisite for M1, M2 and M3 is data completeness within a given area, the other two models require a high spatial accuracy of the input data, mirrored in matching data assignments on individual building levels. Figure 1 shows the workflow of the set-ups of the five models for building value estimation.

**2.2 Intersection with flood hazard maps and spatial aggregation**

Based on the five described models, it is possible to calculate the monetary value of individual buildings (M1, M2, M4 and M5) or mean building values within pre-defined areas (M3). To identify the values, which are exposed to floods, the buildings or areas need to be spatially referenced and overlaid with flood hazard maps. The exposed values based on M3 are defined by the extent of flood exposed areas and their respective monetary value per area. With regard to exposed values

based on individual buildings, we classify a building as *exposed to floods* if it partially or entirely overlaps with a flood-prone area. From this exposed building, the entire monetary value is considered for the calculation of flood exposed values. To compare the  model based on areas (M3) with the other four models, we compile a map of regular hexagons with an area of 10 km$^2$ and calculate the sum of exposed values per hexagon for all five models.

**2.3 Selection of benchmark model and model comparison**

Because our study mainly focuses on comparing different modelling approaches rather than on model predictions, we follow a benchmark test instead of a strict validation procedure. In a first step, we select a benchmark model that best fits with the



direct application of provided portfolio data of Cantonal insurance companies for buildings within eight Swiss Cantons. In a second step, we compare the other four models with the benchmark model and examine the distributions of the extreme high values in more details, including their spatial distributions. In contrast to the selection of the benchmark, the comparison of the benchmark model with the four other models covers the entire modelled area, i.e., the whole of Switzerland.

It is possible to select the model with the best fit in areas, where the data sets of the original building values are complete and spatially referenced on the building level. In our study, these areas correspond to the Cantons, for which complete portfolio data of the Cantonal insurance company for buildings are available, see subsection 2.4.3. Within these Cantons, we attribute the original building values from the portfolio data sets to the corresponding building geometries. Identifying  flood exposed buildings and summing  the exposed values per hexagon are done in the same manner as for the building-based models. To

identify the benchmark model, we examine differences and similarities between the model-based results and the results based on the original building values. For that matter, we calculate the root-mean-square errors (RMSE) and mean absolute errors (MAE) at the data aggregated to hexagons. We compile scatterplots of the hexagon values and compare the sum of exposed values over all hexagons within the validation area. As we are particularly interested in the distribution of the extreme high values, we further fit a generalized Pareto distribution (GPD) to the data above a certain consistent threshold.

The threshold is the location parameter of the GPDs. The other two GPD parameters, the scale and shape, are estimated with the R-package "fExtremes" (Wuertz, 2015) by applying the probability weighted moment method. Furthermore, we compare the highest hexagon values of each data set within the validation area.

## 2.4 Data

Each of the five generic models makes it possible to estimate flood exposed building values based on data sets that are

available in many countries. However, the models' set-up, especially the estimation of the parameter values, requires data sets on monetary building values, which are either representative for a given area (M1 to M3) and/or spatially explicit (M3 to M5). In the following subsection 2.4.1, we present the input data of our study in Switzerland, and in subsection 2.4.2 we detail the data selection for the parameter estimation. Subsection 2.4.3 shortly describes the data and area of model application and comparison.

**2.4.1 Input data**

The main three data sets, which are used for the estimation of the models' parameter values are: (1) points of insurance contracts (PIC), (2) building zone polygons (BZP), and (3) building footprint polygons (BFP). The latter two are also used in the models' application. The PIC data set is a compilation of 552 698 insurance contracts provided by 11 Cantonal insurance companies for buildings (see Fig. 2), harmonized and expressed as values as per 2014. Of these 11 insurance companies,

eight companies provided the whole portfolio data set as per 2013 available, whereas the three remaining companies provided contract data, restricted to contracts with at least one flood claim between 1999 – 2013 (two companies) and 1989 –



2013 (one company), respectively. All data are provided for the exclusive purpose of research and are subject to strict confidentiality.

Cantonal insurance companies for buildings are present in 19 (of totally 26) Swiss Cantons. In these 19 Cantons, the insurance of buildings is compulsory and provided by the respective Cantonal insurance company for buildings, which operates under a legal monopoly. The claims are compensated at replacement costs; thus, the premiums are calculated based on replacement values. Consequently, the portfolio data of a Cantonal insurance company for buildings includes the replacement value of virtually every building within the respective Canton. In addition, most contracts are located on building level - in this study, this is true for 87 % of the provided contracts - and often contain the volume of the insured building or building part. In our case, 78 % of the contracts include this information.

The second input data are the countrywide harmonized BZPs, provided by the Federal Office for Spatial Development (see Table A1 in the appendix A1). For our analysis, we reduce the nine provided building zone categories to six categories by merging the types "restricted building zones", "zones for tourism and sports", and "transport infrastructure within building zones" to the type "other building zones". Furthermore, we add the spatial complement of the building zones as "outside building zone" to the data set. Thus, we obtain a spatially gapless set of polygons with seven different types of building zones; namely, "residential", "working", "mixed", "centre", "public", "others" and "outside building zone".

The third input data are data sets on buildings. In our study, we use the BFP of the swissTLM3D data set, provided by the Federal Office of Topography (see Table A1 in the appendix A1) and harmonized as outlined in Röthlisberger et al. (2017). Three of our building value models consider not only the BFP positions but also various attributes, which we assign to the polygons in preprocessing steps as described in appendix A1. The complete set of attributes considered in the models' set-up consists of six items: (1) building volume above ground, (2) type of building zone and (3) type of municipality within which the BFP is located, (4 and 5) binary information about residential purpose and use, respectively, and (6) building densities in the BFP's surroundings.

The calculation of flood exposed building values does not only require information on building values, but also on flood-prone areas. To define the areas potentially prone to inundation in Switzerland, we combine two different types of flood maps. The main source is a compilation of all available communal flood hazard maps in Switzerland (Borter, 1999; de Moel et al. 2009). These maps are collected, harmonized and provided in agreement with the responsible cantonal authorities by the Swiss Mobiliar company. We use the maps of December 2016, which cover 72 % of the buildings in Switzerland. Out of the five hazard levels indicated in these maps, we consider the levels "high", "medium" and "low" as flood-prone areas. With the selection of these three levels, we include events up to a return period of 300 years. For the 28 % of the buildings in Switzerland that are not covered by the communal flood hazard maps, we use the coarser flood map called Aquaprotect. This data set is provided by the Federal Office for the Environment (Federal Office for the Environment, 2008). Aquaprotect is available for the whole Switzerland and contains four different layers with recurrence periods of 50, 100, 250 and 500 years. For our study, we use the layer with the return period of 250 years. The compilation in GIS of the two map types follows the procedure described by Bernet et al. (2017) and results in a complete, nationwide map of flood-prone areas with return



periods of up to 250 (territories not covered by communal hazard maps) or 300 years (territories covered by communal hazard maps), respectively.

### 2.4.2 Data selection for the parameter estimation

The workflow in Fig. 1 illustrates how the input data are combined and selected for the five models' parameter estimation.

The resulting data selection for each model is summarized in Tab 1.

For M1 to M3, the two countrywide data sets (BFP for M1 and M2, BZP for M3) are reduced to the data entries, which are located within the eight Cantons with complete building insurance data sets. In this way, the BZPs in the set-up of M3 cover 30 % of the data's total coverage, and the number of BFPs used for the parameter estimation of M1 and M2 correspond to 19 % of the total number of BFPs in Switzerland.

The selection of PIC is made in two ways. For the first three models (M1 to M3), we select all PICs within the eight Cantons where complete portfolio data sets are available. For M1 and M2, we directly use the total insured building value of these 529 224 contracts, which corresponds to 412 billion CHF. For M3, however, we further select the PICs that are localized at least at the street level, which is true for 95 % of the PICs in the eight Cantons with complete portfolio data. These PICs are spatially joined with the BZPs within the respective eight cantons. The monetary values of these PIC (400 billion CHF total)

are summarized per BZP type, and the values of the remaining PICs (i.e., 12 million CHF) are split proportionally to the area of each BZP category and added to the respective sum per BZP categories.

For M4 and M5, we reduce the original PIC data provided by 11 insurance companies to the 87 % of points with a localization on building level, and then we assign these points to the nearest BFP with GIS software (see Fig. 1). 92 % of the PICs with a localization on building level can be matched to a BFP within a distance of less than or equal to 5m. The

attributes of these PICs, i.e., the replacement values and volumes of the insured buildings or building parts, are summarized per BFP. With this summation, the BFP with at least one joined PIC contains the attributes of the preprocessing steps (see description in appendix A1), as well as the insurance-sourced building values and volumes. In particular, each of these BFPs includes two types of building volume. The first type is the volume above ground, calculated based on BFP area and the average height above ground of the building during preprocessing of the data. The second type is the sum of volumes

recorded in all PICs, which are assigned to the BFPs. For M4 and M5, we select only those BFPs for which two mentioned volumes are within a pre-defined range. For that matter, we calculate the volume-ratio, i.e., the volume according to PIC divided by the BFP volume above ground. In the eight Cantons, where we obtained complete portfolio data, we consider the volumes as matching if the volume-ratio is equal to or more than 0.8, and less than or equal to 2.0. In the other three Cantons, we set the lower criteria to equal to or more than 1.0. With this comparison of two independently derived volumes,

we efficiently improve the quality of the BFP data. Particularly, we can exclude BFPs with inconsistencies in the calculation of the building volume above ground, and BFPs with mistakenly (not) assigned PICs, which thus have monetary values that are too high (or low). The exclusion of these BFPs is crucial for the set-up of the regression model (M5) and cannot be done manually given the size of the data set. The described comparison of volumes reduces the BFPs and the joined PICs



simultaneously and in a similar way. While 60 % of the BFPs to which a PIC is assigned are finally used for the set-up of M4 and M5, the respective ratio of PICs amounts 59 %.

### 2.4.3 Data and area of model application and comparison

The estimation of the parameter values for all five models is restricted to territories or buildings for which specified building insurance data are available. In contrast to the parameter estimation, applying the models does not require any insurance data and is thus feasible for any territories or buildings with attributes that correspond to the model parameters. In our study, the building referenced models (M1, M2, M4, M5) are applied on the entire BFP data set of 2 086 411 polygons, while M3 is applied to the countrywide BZP data set with an area of 41 290 km$^2$, thus covering all of Switzerland (see Tab 1). The benchmark model is selected in the eight Cantons where complete building insurance data sets are available; for the benchmark test, we again consider the entire territory of Switzerland.

### 3 Results and discussion

In this section, we first show the parameter values of the five building value models, M1 to M5 (subsection 3.1), and then present the results of the benchmark selection and test. The overall discussion of the models in the last subsection (3.4) complements the specific comments in the first three subsections.

### 3.1 Parameter values

**M1 and M2**

The parameter values of the two models with a single, uniform parameter are 1 050 939 CHF per building (M1) and 648.45 CHF/m3 per volume above ground (M2) respectively. These values are rather high compared to international literature data (DEFRA, 2001; Bruijn et al., 2015; Wagenaar et al., 2016), mainly because of comparatively high building standards and construction costs in Switzerland. In addition to and in contrast with  these other studies, we count attached buildings like terraced houses as only one building, and the parameter of M2 refers to the building volume above ground, but includes the costs for underground building volumes too.

**M3 and M4**

Table 2 shows the parameter values of M3, i.e., the monetary values of buildings per surface area [CHF m$^{-2}$] of seven land use categories. Most notable are the value differences between the area inside and outside building zones. The value for the areas outside the building zones is only a very low percentage of the building zones' values, i.e., between 0.3 % (of "centre") and 1.3% (of "others"). Within the building zones, the values show less variation, i.e., they differ by a maximal factor of 4.5 corresponding to the difference between the categories "others" and "centre". Two aspects determine the parameter value of




a specific land use class in M3: firstly, the density (built volume per unit area) of buildings in this land use class and, secondly, the monetary value per built unit volume. The second aspect is at the core of model M4, and the respective parameter values by land use type and building purpose (with or without residential purpose) are presented in Table 3. The monetary value per volume is higher for buildings with residential purpose than for non-residential buildings, ranging

between 17 % for "residential" and "public" building zones to 58 % for areas outside building zones. For residential buildings, the values for different land use types do not vary more than by a factor of 1.9 ("working" to "public"), and by a factor of up to 2.2 for buildings without residential purpose. The ratio between the highest and the lowest M4 parameter value is 2.5. This is the ratio between the value per volume referring to residential buildings in public building zones, and the value per volume, referring to non-residential building outside building zone.

The remarkably smaller variation in parameter values in M4 compared to the variation in M3, and the differences between M3 and M4 in the ranking of land use types by parameter values, all suggest that the differences in building densities have a much higher impact on the variation of M3 parameters than the differences in monetary value per volume. This is especially true for the areas outside building zones, where the M4 values per volume are comparable to the values within building zones. In contrast, the M3 parameter for the area outside building zones is not higher than 1.3 % of the lowest value within

building zones. That low percentage reflects a similarly low ratio between building densities outside and inside building zones. However, the effect of building densities dominates also within building zones. For building zones "centre" and "mixed", the M4 values per volume are at rank four and five, while the M3 parameter values for these zones are at rank one and two. That means the M3 values per area for the building zones "centre" and "mixed" are top ranked, not because of high monetary values per built volume, but because these building zones are densely built-up. In contrast, comparing M3 and M4

parameter values for the zones "public" and "others" suggests that the construction costs for the buildings in these zones are comparably high, but the built volume per area is rather low. In the international literature, the monetary values of buildings per surface area (M3, e.g., Bubeck et al., 2011; ICPR, 2001; Kljin et al., 2007) and the construction costs per building volume (M4, e.g., Arrighi et al., 2013; Fuchs et al., 2015) are remarkably lower than the values in this study. As in the case of M1 and M2, these differences can be explained mainly by differences in building standards and construction costs in

Switzerland. For M3, the relatively dense settlements within building zones in Switzerland are another reason for the comparably high values in our study.

**Regression model M5**

Based on our data, the minimal adequate linear regression model for the estimation of building values is

$$\log_{10}(value) = ResPur \times \log_{10}(volume) + ResPur \times LaUse + \log_{10}(volume) \times LaUse \,, \qquad (1)$$

where *value* is the building value in [CHF], *ResPur* is the binary variable regarding residential purpose (yes/no), *volume* is the building volume above ground in [m$^3$], and *LaUse* is the categorical variable regarding land use (six types of building zones, see subsection 2.4.1). The diagnostic plots of the model are presented in appendix A2 and show that principal

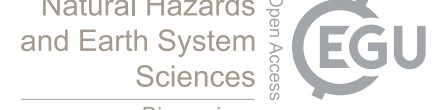



assumptions regarding the residuals are satisfied. The coefficient of determination, adjusted $R^2$, equals 0.88. In other words, M5 explains 88 % of the variance in the logarithmic monetary building values. The overall F-statistic (60 000, on 21 and 172 degrees of freedom) results in a p-value $< 2.2e^{-16}$, indicating an overall significance of the explanatory variables of M5. The estimates of the individual explanatory variables and their pairwise interactions are shown in Table 4, together with standard errors, t- and p-values. With one exception (*ResPur* yes x *LaUse* centre), all parameters of M5 are significant.

The intercept of 3.098 (= 1250 CHF) refers to the variable values of $\log_{10}(volume) = 0$, i.e., *volume* = 1 m$^3$*; ResPur = no* and; *LaUse = outside*. If the same theoretical building of 1 m$^3$ has a residential purpose, the estimation of the monetary value increases by a factor between 4.2 ($10^{(0.793-0.173)}$) in public building zones and 6.2 ($10^{0.793}$) outside building zones or in centre zones. As building volume increases, however, this factor between buildings with and without residential purpose decreases and drops below 1 for building volumes between 8 200 m$^3$ (public building zones) and 102 000 m$^3$ (outside building zones). The effects of land use categories other than "outside" and their interaction with building volumes are similar to the ones of residential purpose, but in the opposite direction. A theoretical building with a volume of 1 m$^3$ in a building zone has a lower building value by factors 0.18 ($10^{-(0.666+0.076)}$, building zones "others", residential building) to 0.39 ($10^{-0.404}$, working zone, no residential purpose) compared to the same building outside building zones. With increasing building volumes, these factors increase and exceed 1 for building volumes between 52 m$^3$ (public building zones, no residential purpose) and 584 m$^3$ (working building zones, residential purpose). In any case, a higher volume of building results in a higher building value, but for all buildings with residential purpose, the increase in value is lower than the increase in volume. Consequently, the ratio of difference in value to difference in volume for residential buildings within the same building zone is below 1. In fact, the ratio ranges from $\Delta volume^{-0.350}$ for areas outside building zones to $\Delta volume^{-0.067}$ for building zone "others". For non-residential buildings, however, the increase in value is higher than the increase in volume in all building zones (with maximal ratio of $\Delta volume^{0.091}$ in building zone "others"), except for building zone "working" ($\Delta value = \Delta volume^{-0.033}$), and for areas outside building zones where the difference in value equals $\Delta volume^{-0.192}$.

In summary, variable values that are different from the intercept generally increase the resulting monetary building values in M5:

- *ResPur*: Buildings with residential purpose have a higher value than non-residential buildings, at least up to a volume of several thousand cubic meters.

- *LaUse:* Buildings in building zones are more expensive than comparable buildings outside building zones, but only if the buildings have a minimal volume of several dozen to a few hundred cubic meters, depending on land use and building purpose.

- *volume*: Higher building volumes result in higher monetary building values, and for non-residential buildings in five building zones (*residential*, *mixed*, *centre*, *public* and *others*) the increase in value is higher than the increase in volume.





The above statement on *ResPur* in M5 is consistent with the relation of residential to non-residential parameter values in M4. M4 and M5 also agree in terms of *LaUse*, apart from building zone "working". However, the findings on the different *Δvolume* to *Δvalue* relations in M5 do not support the concept of a constant value per volume ratio, which is used in M4.

In the following, we summarize the main reasons for excluding originally considered building features (building densities, residential use and municipality types) and for log-transforming the building volumes and values. The features "buildings densities" are all highly correlated with building volume, but they explain less of the building values' variance than the volume (lower adjusted $R^2$, higher AIC). The same holds for residential use with respect to residential purpose. Models that include municipality types and building zones contain many non-significant parameters. Models with municipality types (but without building zones) explain less than corresponding models with building zones (but without municipality types). The building volumes and values are log-transformed since the untransformed values are right skewed and the residuals of models based on untransformed values are heteroscedastic.

## 3.2 Comparison of models with direct application of insurance data for benchmark model selection

The eight Cantons with complete insurance portfolio data cover an area of 12 408 km². The corresponding layer of regular 10 km² hexagons contains 1577 hexagons. Each point in Fig. 3 represents one of these hexagons. The $\log_{10}$ values of flood exposed buildings summarized per hexagon based on value models M1 to M5 (y-axes) are plotted against the exposed $\log_{10}$ values based on the direct application of the values in the spatially referenced building insurance contracts (PIC, x-axis). The red lines indicate a one to one relation. The exposure values per hexagon based on the M2, M4 and M5 models differ hardly more than by a factor of $10^1$ from the respective value based on direct PIC application. Moreover, the factors are homoscedastic. The results from M1 and M3, however, differ by up to a factor of $10^2$ from the ones based on direct insurance data application. In addition, the factors for small values are clearly bigger than the factors for high values. Moreover, in M1 the values of hexagons with only a few exposed buildings are generally overestimated, and the hexagons with one or two exposed buildings appear as two horizontal lines (at $1.05 \cdot 10^6$ and $2.1 \cdot 10^6$ CHF respectively), with only seven hexagons in which the direct application of PIC results in higher exposure values than based on M1. In contrast, the values in hexagons with the most exposed buildings are underestimated in M1. Hexagons with high exposure values are underestimated by the other four value models too, although this is less pronounced in the cases of M2, M4 and M5 than in M1 and M3.

The data in Table 5 support these findings quantitatively. Overall, the indicators for the models M1 and M3 show the least agreement with the values based on directly applied PICs. The sum of exposed values over all 1577 hexagons is closest to the PIC-based result in M5 (+1 %), and the sum differs most in M3 (-29 %). M4 shows the least RMSE and M5 the least MEA; and for both indicators, the values of M1 and M3 are approximately twice as high as the ones of the other three models. Comparing extremely high values shows again a clear division into two groups: M2, M4 and M5 versus M1 and M3. The GPD fitted for hexagons with exposed building values higher than $10^8$ CHF show for M2 and M5 the best match with PIC-based extreme values. The shape parameter determines the weight of the tail in the GPD, and it is highest in case of





direct PIC application, followed by the ones based on M5 (-10.2 %) and M2 (-10.4 %). This general underestimation of extremely high values by the five models is also reflected in the maximal exposure values, where the models result in -25 % (M4) to -72 % (M1) lower values compared to the direct PIC application.

Based on these results, we select M5 as the benchmark model for comparing the countrywide model applications presented in the following section.

## 3.3 Benchmark test: differences and similarities between the five models

The summarized value of all flood exposed buildings in Switzerland is between $3.1 \cdot 10^{11}$ (M3) and $5.4 \cdot 10^{11}$ CHF (M4). Based on the benchmark model M5, it is $4.7 \cdot 10^{11}$ CHF. The ratio between the highest and the lowest sum is thus 1.7, and the ratios to the benchmark model are between 0.7 and 1.1. Table 6 presents the exposure values per eight ranked groups of the total 4444 regular hexagons covering Switzerland. The table demonstrates that for all five models, the distributions of exposed values per hexagon are clearly right-skewed, but for M1 and M3 the skewness is less pronounced. This skew to the right implies that the exposure values of a few $10 \text{ km}^2$ hexagons represent an important part of the total value of flood-exposed buildings in Switzerland. For instance, the 2 % (89) hexagons with the highest exposure values based on M5 contain flood-exposed buildings with a value of $1.6 \cdot 10^{11}$ CHF, which corresponds to 33.6 % of the total value exposed in the whole Switzerland based on M5. This share of exposed values in the 98[th] percentile is comparable for values from M2 (32.8 %) and M4 (34.4 %), but remarkably lower for M1 (23 %) and M3 (28.4 %). Comparing the absolute values of the most exposed hexagons results in the division of the same two clusters, down to the 95[th] percentile, the exposure values based on M2, M4 or M5 are approximately twice as high as the ones based on M1 or M3.

Figure 4 shows the spatial distribution of six ranked groups of hexagons for all five models. The groups' limits in exposed building values per hexagon are presented in columns three to seven in Table 6. The data highlights again the two groups: M2, M4 and M5 versus M1 and M3. However, the spatial distribution of the 1555 (35 %) hexagons with the highest exposure values is very similar with each of the five applied value estimation models. These hexagons cover wide areas in the northern part of Switzerland, but appear as isolated points or lines only in the southern part. Overall, the pattern mirrors the spatial settlement structure (see Fig. 2) in Switzerland, but the areas in the west as well as the most eastern Canton (i.e., GR) seem to exhibit a disproportionally low exposure, which confirms results by Fuchs et al. (2017).

The log-log plots presented in Fig. 5 show the flood exposed values per hexagon based on the benchmark model M5 (x-axis) against the values based on the other four models (y-axes), with the red line indicating a one to one relation. In M2 and M4, the exposed values differ by not more than a factor of five from the respective values based on M5, whereas this factor goes up to $2 \cdot 10^2$ in M3, and to $5 \cdot 10^1$ for M1. In addition, for M1 and M3, the factors are clearly bigger for lower exposure values than for higher ones, and high values in both are generally underestimated. In contrast, the low exposure values in M1 are overestimated, and the values of hexagons with only a few exposed buildings appear as horizontal lines, similar to the pattern shown in the panel M1 of Fig. 3, as discussed above. The M2 panel suggests a general overestimation of the values compared to M5. Moreover, the differences are more pronounced for the middle ranges than for the extreme values. For the





absolute deviations of M4 values from M5, no such dependency from the value's rank can be detected, but the low values are underestimated while high values are overestimated in M4.

Table 7 presents indicators when the M5 benchmark model is compared with the other four models. Overall, these indicators suggest that M2, closely followed by M4, best matches M5. The exposure values based on M1 and M3, though, both agree much less with the M5 results. Compared to M5, M1 and M3 show a general underestimation of flood exposed building values, as well as an underestimation of the extreme high values. In contrast, M4 and, to a lower degree, M2, overestimate the exposure values compared to M5. The parameters of the GPD fitted to hexagons with flood-exposed building values higher than $10^8$ CHF are very similar for M2, M4 and M5. Yet, the resulting empirical cumulative distribution functions presented in Fig.6 for the highest two % show that M2 matches better with M5 than does M4.

## 3.4 Overall discussion of the five models

Based on the resulting values of flood-exposed buildings, the five models can be divided into two groups, one with M1 and M3, and another one with M2, M4 and M5. Compared with the direct application of building values from PIC in eight Cantons, M5 performs best. However, the results based on M2 and M4 are close, too, not only to the PIC results in the eight Cantons (see subsection 3.2), but also to the M5 results over all of Switzerland (see subsection 3.3).

With regard to data requirements for model parameter estimations (see Table1), M5 differs from the other four models, as it is the only model that needs data on individual building level. However, the global sums required in M1 to M4 differ too. While M1 and M2 require relatively simple data, i.e., global values over a particular area such as administrative units, the global sums of monetary building values required for M3 and M4 need to be differentiated to a higher degree. Consequently, the data requirements for the parameter estimation divide the models into three groups, with M1 and M2 in the group with the least requirements and M5 in the one with the most sophisticated requirements. The same grouping occurs when considering the computational expenses of the parameter estimations. While the parameter estimation in M1 and M2 each consists of one numerical division, and of several divisions in M3 and M4 respectively, the set-up of a linear regression model in M5 is an iterative and time-consuming process.

Grouping the models based on data requirements for the model application results in a distinction between M3 and the other four models (see Table 1). Applying M3 requires spatially gapless data on land use, whereas the other four models need information on individual building levels for application. Among these four models, M1 requires the least (location only), while M4 and M5 require the most information about each individual building, i.e., location, volume and other features. With regard to computational expenses for the model application, the five models are similar.

The overall comparison of the five models reveals several things. On the one hand, M5 has the best matching exposure values when compared to the direct application of existing individual building value. On the other hand, M5 requires the most data and computational resources. With M1 and M3, it is the opposite. In summary, all five models have advantages and disadvantages and to select a model means there is a need to balance them. However, selecting a model is often driven



by data availability in real-world applications. As this study shows, selecting a model has consequences for resulting exposure values.

## 4 Conclusions

Our study illustrates the role of building value models into flood exposure and risk analyses at regional to national scales.
With regard to the spatial distribution of exposed building values, the models show widely uniform results. In contrast, the absolute values of exposure differ remarkably. The first finding implies that the spatial prioritization of flood protection measures would be similar with each of the applied value estimation methods. In practice, this means that the application of more sophisticated models does not generally provide a better basis for spatial prioritizations. Consequently, simpler models with lower requirements regarding data input and computational resources are preferable.

The second finding, however, suggests that decision making processes that are based on cost-benefit criteria and, thus, rely on absolute monetary values, are significantly influenced by which building value model one chooses. We find that models based on areas of land use classes, as commonly applied at regional to national scales, underestimate exposure values. The same is true for models based on individual buildings that do not take the building volumes into account. These two model types underestimate the overall exposure, but even more so the extremely high values upon which risk management strategies generally focus. By underestimating exposed values, the protection measures' benefits are underestimated as well. In decision making processes that are based on cost-efficiency, this underestimation would result in suboptimal allocation of resources for protection measures. Consequently, we propose that estimating exposed building values should be based on individual buildings rather than on areas of land use types. In addition, the buildings' individual volume has to be taken into account in order to provide a reliable basis for cost-benefit analyses. The consideration of other building features further improves the value estimation.

In our study for the whole Switzerland, with a data aggregation on 10 km$^2$ hexagons, the optimal model for the estimation of absolute monetary building value is M5, i.e., a linear regression model considering the residential purpose and the building zone, in addition to buildings' volumes. In other contexts, where other data with different aggregations are available, the optimal building value model may be another one. For decisions that rely on absolute monetary building values, however, our results suggest using a value model based on individual building data that in any case includes the building volume. The concepts of the three respective value models presented in this study, i.e. M2, M4, and M5, are generic. Thus, these models are transferrable with minimal adjustments according to the application's purpose and the available data. However, within the context of flood risk management, the optimal value estimation model depends on the specific questions to be answered.





**Appendix A1**

**Details on data and assignment of attributes to building polygons**

Table A1 presents details on the data sets, which we use in our study aside from the insurance data described in subsection 2.4. We assign the attributes to the building footprint polygons as follows.

**Building volume above ground**

The building volume above ground is the product of the BFP area times the buildings' average height above ground. While the calculation of a polygon's area is a standard procedure in GIS, the estimation of the building height based on the available data is a multistep process. First, the points of the digital elevation model (swissALTI3D) and the digital surface model (DSM) are assigned to the polygons and for each polygon the two means of the assigned swissALTI3D points and

DSM points respectively, are calculated. The subtraction of the mean of the DSM points from mean from the swissALTI3D points results in the building's average height above ground. If this height is ≥ 3.5 m and ≤ 100 m (which is the case for 1 378 665 of total 2 086 411 BFPs) it is used in the volume calculation, otherwise (n = 707 746) it is adjusted as follows: For residential buildings (i.e., buildings with assigned residential units as explained further down, n = 232 016) the average numbers of floors of the assigned BDS points (attribute GASTWS in BDS) is calculated and for the first floor the height is

set to 3.5 m and for each additional floor 2.5 m are added. For non-residential buildings with a height < 3.5 m or > 100 m (n = 475 730) the value is set to 3.5 m.

**Type of building zone and type of municipality**

For the assignment of the types of building zones and municipalities respectively, the positions of the building polygons' centroids relative to the polygons in the data sets "Bauzonen Schweiz" and INFOFLAN-ARE respectively are analysed.

Prior to the assignment, we reduce in our study the types of building zones (attribute "CH_BEZ_D" in the data set "Bauzonen Schweiz") from nine to seven types as described in subsection 2.4.1. The types of municipalities (attribute TYP in INFOPLAN-ARE) are reduced from originally nine types down to six by merging the types "big centres" (code "1" in TYP), "secondary centres beside big centres" (2), and "middle centres" (4) to the type "big and middle centres" and by merging "belts of big centres" (3) and "belts of middle centres" (4) to the type "belts of big and middle centres".

Furthermore, we add the areas of lakes, if they are not part of a municipality but of a Canton, to the type "agricultural" (code "8" in TYP) municipality. We obtain a spatially gapless set of polygons with six types of municipality, namely "big and middle centres", "belts of big and middle centres", "small centres", "suburban rural municipalities", "agricultural municipalities and cantonal lake areas", and "tourist municipalities".





**Binary information about residential purpose and use**

The point data of residential units in the BDS (n = 1 670 540) are joint to the next BFP (n = 2 086 411) within 2 m. 97 % (1 631 531) of the BDS points lay in or within a distance of 2 m to a BFP. We consider a BFP as a building with residential purpose, if at least one BDS point is assigned to it (n = 1 269 908 BFPs.) The criteria for residential use is that at least one person with main residence (attribute GAPHW in the BDS data set) is assigned to the building polygon, which is true for 1 129 904 BFPs.

**Building densities in the BFPs' surroundings**

For the calculation of the building density in the surrounding of a BFP we define circles of 50, 100, 200 and, 500 m radius, around the BFP's centroid. For each of these circles we calculate the area of all BFP (cut to the circle's edge) and divide it by the total area of the circle (cut to areas within Switzerland and not covered by lakes). This way, we obtain for each BFP the building density in a circle 50 m (100, 200 and 500 m) around its centroid.

**Appendix A2**

**Diagnostic plots of linear regression model M5**

Figure A2 shows the diagnostic plots of M5, the minimal adequate linear regression model presented in subsection 3.1. The two plots of residuals versus fitted values suggest (Fig. A2 a and Fig. A2 c) that residuals fulfill the assumptions of homoscedasticity, as the residuals are spread equally along the ranges of the fitted values. The quantile-quantile plot (Fig. A2 b) indicates, that the tales of the residuals' distribution are heavier than in a normal distribution. The cook's distance plot (Fig. A2 d) shows that all buildings are inside Cook's distance of 0.5, that means, no building influences significantly the resulting regression model. Overall it can be stated, that the principal assumptions of linear regression modelling are reasonably satisfied.

**Competing interests**

The last author is a member of the editorial board of the journal. Otherwise, the authors declare that they have no conflict of interest.

**Acknowledgments**

Funding from the Swiss Mobiliar ('die Mobiliar AG') supported the completion of this research. We thank the natural hazards group at Swiss Mobiliar for acquiring and compiling the communal flood hazard maps. Furthermore, we would like



to thank the public insurance companies for buildings of the cantons Aargau, Basel-Landschaft, Fribourg, Glarus, Graubünden, Jura, Neuchâtel, Nidwalden, St Gall, Solothurn and Zug for providing building specific data and supporting us during the data harmonization process. Also, we would like to thank the Federal Office of Topography, the Federal Office for Spatial Development, the Federal Office for the Environment and the Federal Statistical Office for providing the

corresponding spatial data. Last but not least, we thank Daniel Bernet for the joint effort to collect and harmonize the insurance data and for proofreading, Markus Mosimann for his support harmonizing the insurance data and Craig Hamilton for language editing..

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



**Tables and Figures**

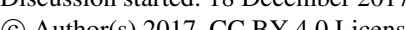

Figure 1: Workflow of the set-ups of the five investigated models for building value estimations.




**Cantonal insurance companies that provided data**

AG: Aargau (1989 - 2013)
BL: Basel-Landschaft (1999-2013)
FR: Fribourg
GL: Glarus
GR: Graubünden
JU: Jura
NE: Neuchâtel
SG: St. Gallen (1999-2013)
SO: Solothurn
ZG: Zug

**Data availability**

Complete data set of insurance contracts

Incomplete data set of insurance contracts

**Figure 2: Overview of provided data by the Cantonal insurance companies for buildings. Three insurance companies only provided data limited to contracts associated with at least one flood claim within the period indicated in brackets. The grey shaded area indicate the footprints of all buildings in Switzerland.**



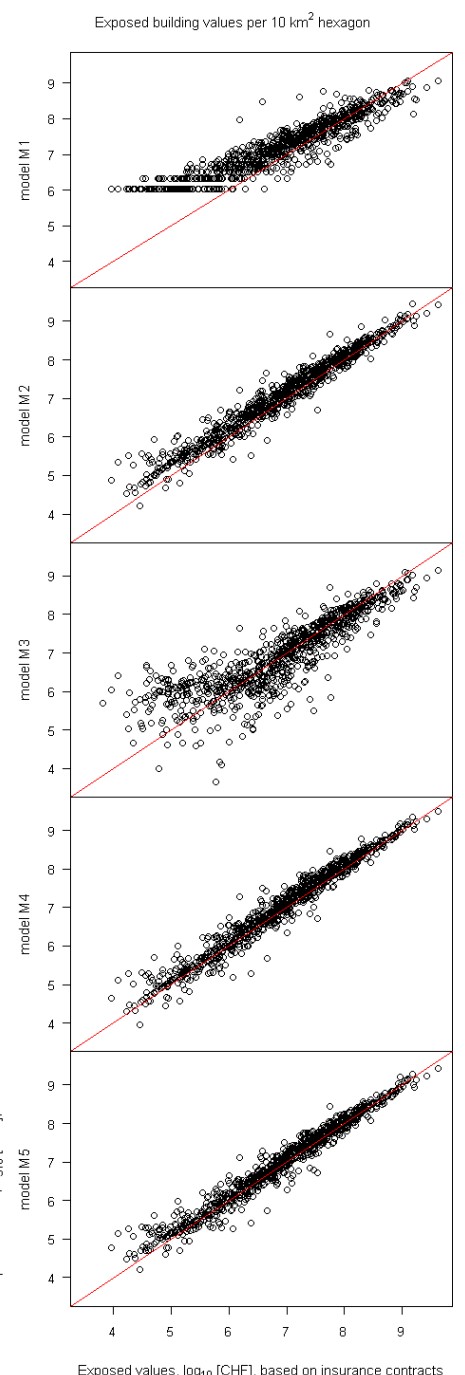

**Figure 3: Scatterplot of flood exposed buildings' values, aggregated to regular hexagons with a surface area of 10 km$^2$. The sums based on models M1 to M5 (y-axis) are plotted against the sums based on the direct application of the values from the spatially referenced building insurance contracts (x-axis). The red lines indicate the 1:1 relation. The values are log$_{10}$ transformed and sums below 10$^4$ CHF are not shown.**





**Figure 4: Spatial distribution of flood exposed building values based on benchmark model M5 (uppermost figure), in addition tomodels M1 to M4 (lower figures). Hexagons with a surface area of 10 km² are categorized according to their sum of flood exposed building values. The specific limits of each category and the corresponding sums of exposed values are presented in Table 6.**

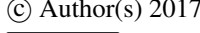



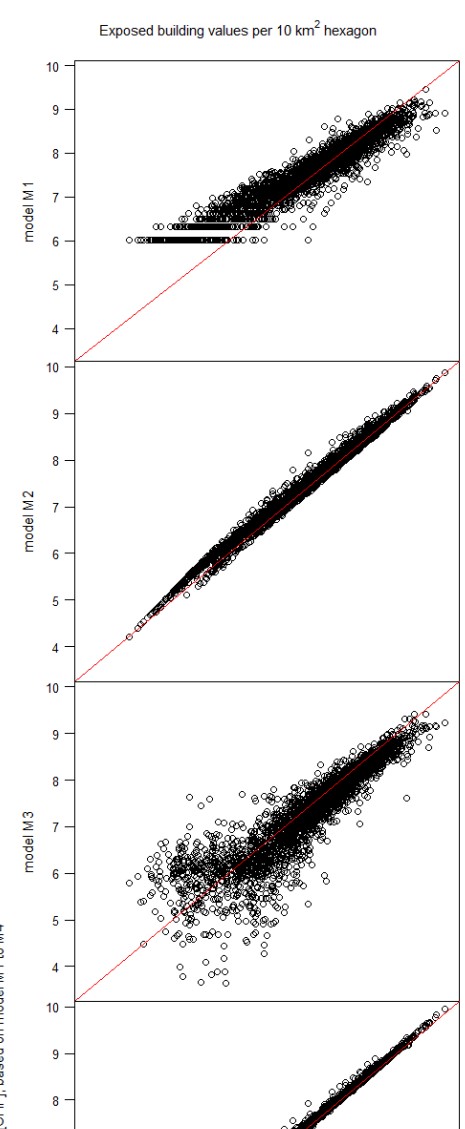

**Figure 5: Scatterplot of flood exposed buildings' values, aggregated to regular hexagons with a surface area of 10 km². The sums based on models M1 to M4 (y-axis) are plotted against the sums based on the benchmark model M5 (x-axis). The red lines indicate the 1:1 relation. All values are log₁₀ transformed and sums below 10⁴ CHF are not shown.**





**Figure 6: Empirical cumulative distribution function of flood exposed building values aggregated to hexagons with a surface area of 10 km². Cumulative probabilities (p) are generated by $10^5$ random values from GPD with the parameters shown in Table 7. To improve the readability, only probabilities over 98 % are shown.**





| Model name and concept | M 1 uniform average value per building | M 2 uniform average value per building volume | M 3 average value of buildings per area, differentiated by land use category | M 4 average value per building volume, differentiated by building features | M 5 value per building, individually calculated based on linear regression |
|---|---|---|---|---|---|
| **Parameter estimation and [unit]** | Total value of buildings in an area divided by total number of buildings in the same area, [CHF] | Total volume of buildings in an area divided by total number of buildings in the same area, [CHF/m$^3$] | Total value of buildings within an area of a particular land use category divided by the size of the area, [CHF/m$^2$] | Total value of buildings with identical features divided by the volume of the buildings, [CHF/m$^3$] | Minimal adequate linear function of building features, [CHF] |
| **Data for parameter estimation** | | | | | |
| - minimal requirement | Global sums of values and numbers of buildings within a given area | Global sums of values and volumes of buildings within a given area | Global sum of building values within an area with particular land use size of the area | Global sums of values and volumes of buildings with identical features | Individual values and features of buildings |
| - used in this study | Complete data of eight Cantons where entire portfolio insurance data are available: | | | BFP of eleven Cantons, reduced to polygons with joined PIC and matching volumes (n =172 562) | |
| | - Total of insured buildings values in 529 224 PIC | - Total of insured buildings values in 529 224 PIC | - Total of insured buildings values in 529 224 PIC | BFP including volume, summarized value of joined PIC and information on land use and building purpose | BFP including volume, summarized value of joined PIC and information on land use, municipality type, building purpose and use |
| | - Total number of BFP (391 766) | - Total volume of BFP (653×10$^6$m$^3$) | - BZP of 12 408 km$^2$, covering the entire area | | |
| **Data for benchmark selection** | The data must be spatially referenced at object level and complete within a given area. In this study, we use the 529 224 PIC of the eight cantons, where complete portfolio data of the Cantonal insurance company for buildings are available. | | | | |
| **Data for model application** | | | | | |
| - minimal requirement | Individual buildings: location only | Individual buildings: location and volume | Land use: spatially gapless information on land use categories | Individual buildings: location, volume and features | |
| - used in this study | BFP data set of 2 086 411 footprints | BFP data set of 2 086 411 footprints, including volume | BZP of 41 290 km$^2$, covering entire Switzerland | BFP data set of 2 086 411 footprints, including volume and information on land use and building purpose | |
| **Frequent fields of applications** | Default values in tools for cost-benefit analyses of flood protection measures | | Widely used in flood risk analyses at regional to nat. scales | Mainly used in construction industry and real estate management for the estimation of individual building construction costs | |
| Examples | DEFRA, 2001; Wagenaar et al., 2016; van Dyck and Willems, 2013 | BAFU, 2015; de Bruijn et al., 2015; Mobiliar Lab, 2016; Winter et al., 2017 | Bubeck et al., 2011; Cammerer et al. 2013; ICPR, 2001; Klijn et al., 2007; Thieken at al., 2008 | Hägi, 1961; Naegeli and Wenger, 1997; SVKG and SEK/SVIT, 2002. Few applications in flood risk management, mainly at local | Lowe et al., 2006; Sonmez, 2008. To our knowledge no application in flood risk management |



level  e.g., Arrighi et
al. 2013,

**Table 1: Overview on concepts, data and application of the five investigated models for building values estimation. BFP stands for building footprint polygons, BZP for building zone polygons and PIC for points of insurance contracts.**




| Type of land use | Area [10³ m²] | Value [10³ CHF] | Value per area, directly assigned [CHF/m²] | Value per area, total [CHF/m²] |
|---|---|---|---|---|
| centre | 64 974 | 95 118 671 | 1 463.94 | **1 464.88** |
| mixed | 25 705 | 28 915 610 | 1 124.89 | **1 125.84** |
| residential | 182 593 | 163 193 242 | 893.76 | **894.70** |
| public | 47 323 | 30 448 192 | 643.42 | **644.36** |
| working | 57 652 | 33 982 269 | 589.44 | **590.38** |
| others | 25 593 | 8 493 504 | 331.87 | **332.81** |
| outside building zone | 12 003 959 | 39 851 666 | 3.32 | **4.26** |
| not directly assigned | -- | 11 719 182 | -- | 0.94 |
| total | 12 407 798 | 411 722 336 | 32.24 | 33.18 |

Table 2: Parameter values of the model M3, value per surface area [CHF/m²] of seven types of land use, i.e., six types of building zones and the area outside of building zones, based on complete portfolio data of eight Cantonal insurance companies for buildings in Switzerland. Insured values of buildings, which are localized at least on street level, are directly assigned to a type of land use. Values of the remaining buildings are split over all types of land use according to the size of the area of each type. The results per type of land use, which are used for the further analyses, are in bold. Table entries are ordered by rank of these results.

| Type of land use | With residential purpose | | | Without residential purpose | | |
|---|---|---|---|---|---|---|
| | Insured building values [10³ CHF] | Volume of buildings [10³ m³] | Value per building volume [CHF/m³] | Insured building values [10³CHF] | Volume of buildings [m³] | Value per building volume [CHF/m³] |
| public | 9 684 445 | 10 150 | 954 | 7 068 467 | 8 640 | 818 |
| others | 2 322 506 | 2 446 | 950 | 866 757 | 1 187 | 730 |
| residential | 110 421 355 | 123 056 | 897 | 2 263 843 | 2 960 | 765 |
| centre | 56 405 627 | 65 486 | 861 | 3 452 311 | 5 351 | 645 |
| mixed | 15 792 658 | 19 708 | 801 | 3 107 297 | 5 321 | 584 |
| outside building zone | 9 668 384 | 16 221 | 596 | 4 908 676 | 13 062 | 376 |
| working | 7 702 381 | 15 259 | 505 | 12 140 152 | 32 234 | 377 |

Table 3: Parameter values of model M4, value per volume [CHF/m³] above ground, differentiated according to the area's land use where each building is located and by the purpose of the building. Calculations are based on insured values of 172 562 buildings, which are provided by eleven cantonal insurance companies in Switzerland. Table entries are ordered by the value per building volume of buildings with residential purpose.





| Parameter | Estimate | Standard error | t-value | Pr(>\|t\|) |
|---|---|---|---|---|
| *Intercept* | 3.097512 | 0.00633 | 489.334 | < 2.00E-16 |
| *ResPur* yes | 0.793809 | 0.007992 | 99.323 | < 2.00E-16 |
| $\log_{10}(volume)$ | 0.80819 | 0.002385 | 338.9 | < 2.00E-16 |
| *LaUse* residential | -0.51207 | 0.009017 | -56.79 | < 2.00E-16 |
| *LaUse* working | -0.4035 | 0.016537 | -24.4 | < 2.00E-16 |
| *LaUse* mixed | -0.65351 | 0.015906 | -41.087 | < 2.00E-16 |
| *LaUse* centre | -0.70887 | 0.009651 | -73.453 | < 2.00E-16 |
| *LaUse* public | -0.44107 | 0.017177 | -25.678 | < 2.00E-16 |
| *LaUse* others | -0.6658 | 0.027504 | -24.208 | < 2.00E-16 |
| *ResPur* yes x $\log_{10}(volume)$ | -0.15846 | 0.002694 | -58.815 | < 2.00E-16 |
| *ResPur* yes x *LaUse* residential | -0.14691 | 0.003563 | -41.23 | < 2.00E-16 |
| *ResPur* yes x *LaUse* working | -0.03614 | 0.005837 | -6.192 | 5.95E-10 |
| *ResPur* yes x *LaUse* mixed | -0.05128 | 0.005654 | -9.071 | < 2.00E-16 |
| *ResPur* yes x *LaUse* centre | -0.0001 | 0.003439 | -0.029 | 0.977 |
| *ResPur* yes x *LaUse* public | -0.17378 | 0.006391 | -27.19 | < 2.00E-16 |
| *ResPur* yes x *LaUse* others | -0.07611 | 0.011406 | -6.673 | 2.52E-11 |
| $\log_{10}(volume)$ x *LaUse* residential | 0.258569 | 0.003217 | 80.371 | < 2.00E-16 |
| $\log_{10}(volume)$ x *LaUse* working | 0.158917 | 0.004704 | 33.787 | < 2.00E-16 |
| $\log_{10}(volume)$ x *LaUse* mixed | 0.26366 | 0.004834 | 54.542 | < 2.00E-16 |
| $\log_{10}(volume)$ x *LaUse* centre | 0.263382 | 0.003448 | 76.397 | < 2.00E-16 |
| $\log_{10}(volume)$ x *LaUse* public | 0.256911 | 0.005323 | 48.262 | < 2.00E-16 |
| $\log_{10}(volume)$ x *LaUse* others | 0.282637 | 0.009382 | 30.127 | < 2.00E-16 |

**Table 4: Parameter estimates, standard errors, t- an p-values of the three explanatory variables (and their pairwise interaction) of model M5. The three explanatory variables are residential purpose (*ResPur*) with levels "yes" and "no", the building volume above ground in m$^3$ (*volume*) and land use (*LaUse*) with levels "residential", "working", "mixed", "centre", "public", "others" and "outside" (i.e., area outside building zones). The *intercept* stands for the variable values of $\log_{10}(volume) = 0$, (i.e., volume = $1 m^3$); $ResPur = no$ and $LaUse = outside.$**





| Method | Comparison over 1 577 hexagons | | | Comparison of extreme values | | |
|---|---|---|---|---|---|---|
| | | | | Fitted GPD for hexagons > $10^8$ [CHF] | | |
| | SUM [$10^6$ CHF] | RMSE [$10^6$ CHF] | MAE [$10^6$ CHF] | SHAPE | SCALE [$10^6$] | MAX [$10^6$ CHF] |
| M 1 | 55 667 | 124 | 23 | 0.25639 | 128 | 1 163 |
| M 2 | 74 451 | 73 | 14 | 0.44574 | **151** | 2 874 |
| M 3 | 47 880 | 115 | 21 | 0.31655 | 117 | 1 367 |
| M 4 | 76 956 | **52** | 12 | 0.43464 | 162 | **3 127** |
| M 5 | **68 111** | 60 | **11** | **0.44709** | 143 | 2 682 |
| PIC | 67 375 | -- | -- | 0.49797 | 149 | 4 157 |

**Table 5:** Indicators for the comparison of model M1 to M5 with the direct application of insurance data (insured values according to point-referenced building insurance contracts, PIC), in the eight Cantons where complete portfolio data of the Cantonal insurance companies for building are available. SUM represents the sum of exposed building values over all 10 km² hexagons, RMSE and MAE represent the root-mean-square error and the mean-absolute error of exposed building values per hexagon when comparing M1 to M5 with PIC. The generalized Pareto distribution (GPD) is fitted for hexagons with exposed building values higher than $10^8$ CHF, which is equal to the location parameter of the GPD. SHAPE and SCALE represent the respective parameter of the fitted GPD. MAX represents the highest sum of exposed building values per hexagon. Bold numbers indicate the value (of M1 - M5) nearest to the value based on PIC.

| Hexagon group | | Lower limit [$10^6$ CHF] of exposed building values per hexagon | | | | | Monetary value of exposed buildings per hexagon group: sum [$10^9$ CHF] and percentage [%] of total | | | | | | | | | |
|---|---|---|---|---|---|---|---|---|---|---|---|---|---|---|---|---|
| *Share [%]* | *Number [#]* | M 1 | M 2 | M 3 | M 4 | M 5 | M 1 | | M 2 | | M 3 | | M 4 | | M 5 | |
| | | | | | | | S* | *P* | S* | *P* | S* | *P* | S* | *P* | S* | *P* |
| *1* | 44 | 739 | 1 518 | 827 | 1 545 | 1 409 | 48 | *14.1* | 112 | *21.9* | 57 | *18.4* | 127 | *23.6* | 107 | *22.8* |
| *2* | 89 | 590 | 1 057 | 585 | 1 114 | 980 | 77 | *23.0* | 168 | *32.8* | 88 | *28.4* | 185 | *34.4* | 158 | *33.6* |
| *5* | 222 | 353 | 550 | 344 | 565 | 500 | 137 | *40.8* | 268 | *52.4* | 146 | *47.0* | 287 | *53.5* | 248 | *52.8* |
| *10* | 444 | 224 | 303 | 197 | 306 | 274 | 200 | *59.4* | 358 | *70.1* | 204 | *65.7* | 380 | *70.7* | 330 | *70.2* |
| *20* | 889 | 108 | 129 | 88 | 134 | 119 | 270 | *80.3* | 447 | *87.3* | 264 | *85.2* | 471 | *87.8* | 412 | *87.5* |
| *35* | 1 555 | 41 | 38 | 27 | 38 | 34 | 317 | *94.1* | 496 | *97.0* | 299 | *96.5* | 523 | *97.3* | 457 | *97.2* |
| *50* | 2 222 | 12 | 7 | 5 | 6 | 6 | 333 | *99.0* | 509 | *99.6* | 308 | *99.5* | 536 | *99.7* | 469 | *99.7* |
| *100* | 4 444 | 0 | 0 | 0 | 0 | 0 | 336 | *100* | 511 | *100* | 310 | *100* | 537 | *100* | 470 | *100* |

**Table 6:** Hexagons of 10 km² grouped in decreasing order of monetary values of flood-exposed buildings in Switzerland. For each group of hexagons and each model (M1 to M5) the following entities are reported: the lower limit of exposed building values per hexagon (in $10^6$ CHF), the sum (S* in $10^9$ CHF) of exposed building values over all hexagons of the respective group, and the percentage (P* in %) of this sum per group in relation to the total value of flood exposed buildings in Switzerland. The spatial distribution of six of these groups (highest 2 %, lowest 65 % and four groups in between) are shown in Fig. 4.





| Model | Comparison over 4 444 hexagons | | | Comparison of extreme values | | |
|---|---|---|---|---|---|---|
| | | | | Fitted GPD for hexagons > $10^8$ [CHF] | | |
| | SUM [$10^6$ CHF] | RMSE [$10^6$ CHF] | MAE [$10^6$ CHF] | SHAPE | SCALE [$10^6$] | MAX [$10^6$ CHF] |
| M 1 | 336 460 | 214 | 47 | 0.20147 | 155 | 2 912 |
| M 2 | 511 208 | 52 | 15 | 0.41285 | 207 | 7 546 |
| M 3 | 309 794 | 191 | 44 | 0.30666 | 148 | 2 634 |
| M 4 | 536 989 | 65 | 15 | 0.43357 | 211 | 9 102 |
| M 5 | 470 420 | -- | -- | 0.42715 | 188 | 7 201 |

**Table 7: Indicators for the comparison of model M1 to M4 with benchmark model M5. SUM represents the sum of exposed building values over all hexagons, RMSE and MAE represent the root-mean-square error and the mean-absolute error of exposed building values per hexagon when comparing M1 to M4 with M5. The generalized Pareto distribution (GPD) is fitted for hexagons with exposed building values higher than $10^8$ [CHF], which is equal to the location parameter of the GPD. SHAPE and SCALE**
10 **represent the respective parameter of the fitted GPD. MAX represents the highest sum of exposed building values per hexagon.**



| Name | Consideration in model | | Data set | Description | Source |
| | set-up | application | | | |
|---|---|---|---|---|---|
| Building footprints BFP | M1, M2, M4, M5 | M1, M2, M4, M5 | swissTLM3D | Feature TLM_GEBAEUDE_FOOTRPINT of the Swiss topographical landscape model, v1.4, as of 2016 | Federal Office of Topography (swisstopo) *https://shop.swisstopo.admin.ch/ en/products/landscape/tlm3D* |
| Polygons of building zones BZP | M2, M3, M4, M5 | M2, M3, M4, M5 | Bauzonen Schweiz (harmonisiert) | Polygons of building zones, 9 harmonized types, as of 2012 | Federal Office for Spatial Development (ARE) *http://www.kkgeo.ch/ geodatenangebot/ geodaten-bauzonen-schweiz.html* |
| Digital elevation model | M2, M4, M5 | M2, M4, M5 | swissALT3D | High precision digital elevation model of Switzerland, grid size of $2 \times 2$ m, as of 2013 | swisstopo *https://shop.swisstopo. admin.ch/en/ products/ height_models/ alti3D* |
| Digital surface model | M2, M4, M5 | M2, M4, M5 | DSM | Digital surface model, density of 1 point per 2 m$^2$, last updated in 2008 | swisstopo *https://shop.swisstopo. admin.ch/en/ products/ height_models/ DOM* |
| Munici-pality types | M4, M5 | | INFOPLAN-ARE | Typology of municipalities ARE, 9 types based on municipality typology of FSO, as of 2014 | ARE, Federal Statistical Office (FSO) and swisstopo *data.geo.admin.ch/ ch.are.gemeindetypen/ data.zip* |
| Residential purpose of buildings | M4, M5 | M4, M5 | BDS | # of residential units in the Buildings and Dwellings statistics BDS, as of 2012 | FSO *https://www.bfs.admin.ch/ bfs/en/home/statistics/ construction-housing/ surveys/gws2009. assetdetail.8521.html* |
| Residential use of buildings | M4, M5 | | BDS | # of people with main residence in BDS, see residential purpose | see residential purpose |
| Area of Cantons | M1, M2, M3 | | SwissBOUN-DARIES3D | Polygons of the 26 Swiss Cantons (districts), as of 2016 | swisstopo *https://shop.swisstopo. admin.ch/en/ products/landscape/ boundaries3D* |

**Table A1: Summary of data used in the set-up and/or application of the five building value models. All links were last checked on 15 September 2017.**







**Figure A2: Diagnostic plots of model M5, namely residuals vs. fitted values (a); quantile-quantile plots of residuals vs. normally distributed quantiles (b); scale location plot (c) and; cook's distance plot (d).**