# Peer review of "A Comparison of building value models for flood risk analysis"

_Natural Hazards and Earth System Sciences, 2017_

## Referee Comment (RC1) · Anonymous Referee #1 · 23 Jan 2018

Dear authors, thank you very much for this interesting contribution to NHESS. The paper addresses the problem of how to estimate asset values of buildings, which can be further used as input for exposure analyses and finally for (flood) risk analyses. Exposure analysis is often neglected in the scientific literature. Therefore, this detailed investigation is highly appreciated.

Taking Switzerland as an example, five models to estimate building values were derived and compared to a comprehensive set of insurance data as well as to a benchmark model (model 5). The five models require different data inputs ranging from simple information, e.g. average value per building or per unit of a specific land use category, to data containing different features (e.g. volume) of individual buildings.

The investigation is straightforward and the findings and conclusions are quite clear.

[Figure]

Since all five models reveal similar spatial pattern of asset values throughout Switzerland, a simple model, e.g. based on land use units, can be used for a countrywide screening of protection needs or prioritisations of regions. However, when it comes to a quantification of costs and benefits of risk reduction measures simple approaches tend to underestimate asset values and therefore project benefits. Therefore, such estimations should preferably be based on individual buildings.

While the paper is in general clearly written and presented, I would like to raise to a few topics of concern:

1) In the introduction, a more general discussion about valuation methods and their application fields is missing. Commonly, approaches based on replacement values are distinguished from approaches that rely on depreciated values. Insurance values or market values can be used to approximate one or the other. A distinction of these approaches and their application fields (e.g. insurance claims, cost-benefit-analysis) should be added. This issue should also be reflected later in the discussion. In the current paper, this issue is only very briefly mentioned in section 2.4.1, which is late and not sufficient in depth.

2) The five models are well explained, but the rationales/justifications behind these models remain unclear. Please add some more background and assumptions about all models. Tab. 1 provides a comprehensive overview, but needs in my view more explanation in the main text. The same holds for Fig. 1.

3) The authors find big differences between the Swiss unit costs and other published unit costs and explain this by differences in building standards and higher construction costs in Switzerland. It would be helpful to add some additional (real) data or statistics that underpin this explanation.

4) An overview table with advantages and disadvantages of all five models and their suitable applications would be helpful to summarize the findings (as a kind of counterpart to Table 1).

5) The conclusion should end with an outlook on future research perspectives. (The implications of the results are addressed sufficiently.)

A few minor comments are:

1) In the introduction, a few important papers in this field are missing from my point of view, e.g.:

- Barredo, J. I. (2009): Normalised flood losses in Europe: 1970–2006. - Nat. Hazards Earth Syst. Sci. 9: 97-104.

- Jongman, B., Koks, E. E., Husby, T. G., and Ward, P. J. (2014): Increasing flood exposure in the Netherlands: implications for risk financing. - Nat. Hazards Earth Syst. Sci. 14: 1245-1255.

- Kleist, L., A.H. Thieken, P. Köhler, M. Müller, I. Seifert, D. Borst, U. Werner (2006): Estimation of the regional stock of residential buildings as a basis for comparative risk assessment for Germany. – Nat. Hazards Earth Syst. Sci. 6: 541-552.

- Seifert, I., A.H. Thieken, M. Merz, D. Borst, U. Werner (2010): Estimation of industrial and commercial assets values for hazard risk assessment. – Natural Hazards 52: 453-479.

2) Instead of "annual expected loss" either "expected annual damage" (EAD) or "average annual loss" (AAL) should be used.

3) All abbreviations should be explained in the text once (e.g. AIC).

4) The hazard levels (high, medium, low) should be explained for readers who are not familiar with the Swiss hazard zones.

5) The correctness of the terms "global sums" or "global values" in section 3.4 should be checked. These sound a bit weird in this context.

I am looking forward to the revised version.

---

## Referee Comment (RC2) · Anonymous Referee #2 · 26 Feb 2018

The manuscript "On the role of building value models for flood risk analysis" describes five different models to estimate the exposure value. It compares the exposure value estimates with insurance data about exposure values and then compares the models too each other. It concludes that models based on land-use types and models based on uniform building values don't perform well compared to models that take into account building volume. The study seems to be carried out well and this exact topic is to my knowledge not published about before. Reading the paper raised however several questions and requests for clarification (see below).

More important points:

- The title is interesting, I like it. However I feel like the connection between the content of the paper and the title isn't very good yet. The role of building value models in flood

risk analysis gets very little attention compared to building value models in general. Could you either come up with a new title or reframe the abstract, introduction and conclusion a bit so the link between the paper and the title is better.

- On Page 9, line 20 you state that attached buildings are counted as one building, this is quite different from other models that I know. Maybe this is why you find the extreme values so important? I can imagine uniform models perform really bad because of this. This also makes the building stock of a particular area very important for the model performance with a uniform value. Maybe discuss this assumption behind the paper a bit better because it could make the results not applicable to other areas.

- If I understand it correctly you first determined which model performs best for the places that have good data available. Then you compare the other models to this best "benchmark" model for the whole country. Why do you take this two-step approach? Why not just compare everything to the available insurance data points and make your judgment on this?

- How accurate is the insurance data that the benchmark model was picked on? Are these values also based on a model or are these expert estimates? If expert estimates, do these experts have some valuation model that they apply? My worry is that the insurance data has artificial relationships in it (based on their valuation model) and that this study is just recreating the valuation model of the insurance companies. In that case the model that is closest to the currently applied insurance models performs best. Can you please discuss to what extend this is a possibility?

- The conclusion states that M1 and M3 underestimate the exposure values. This sounds like a bias in the model. Is this bias not simply a problem with the parameter values rather than a problem with the model itself? If you would just increase the parameters values wouldn't that get rid of the entire bias? Or I think I probably misunderstood what you meant by this sentence so please try to explain this a bit better.

- In this paper a clear separation is made between exposure and vulnerability. This is

common in the literature. However, exposure values can also be combined with the vulnerability in flood risk analysis, for example in an absolute damage functions. This has the advantage that you only need one model rather than an exposure value model and then a model to estimate the damage fraction to be multiplied by this exposure value. An example of this setup is Wagenaar et al. (2017) which used Machine Learning methods and damage data to directly estimate the damage. These estimates are based on both hazard and exposure characteristics. Implicitly such models therefore also include an exposure valuation model. Could you discuss the benefits of the two step approach taken in this study (exposure value separated from vulnerability)?

Minor points:

- Page 1, line 24. Please explain how risk management focuses on extreme exposure values. I'm a bit skeptical about this so please convince me.

- Page 5, line 28. Please explain what a hexagon is in this context.

- Page 12, line 19. Please explain homoscedastic.

- Figure 1 and the data selection section (2.4.2) are currently difficult to understand. Please start with why data selection is required and then lead the reader along figure 1 explaining at every step why each action is carried out.

- Establish a clear definition of building value in the introduction. You mention it is replacement value later on but maybe move that to the introduction. Also a good definition of replacement value is important and useful (can also be a reference). This seems quite relevant for this paper.

- In section 2.2 please explain that in your definition of exposure only buildings are included that can actually flood. I also know definitions of exposure in which any building is included.

- In 2.4.1 a whole list of abbreviations is introduced at once. This is difficult to follow and makes the text a bit of a puzzle (especially on the first read). So I would choose

not to use abbreviations in this case.

- Page 6, line 12-18. Please explain better why this approach is required. I don't get why you need the entire benchmarking next to the validation on insurance data.

- Page 12, line 30. MEA is I think MAE

Reference:

Wagenaar, D., de Jong, J., and Bouwer, L. M.: Multi-variable flood damage modelling with limited data using supervised learning approaches, Nat. Hazards Earth Syst. Sci., 17, 1683-1696, https://doi.org/10.5194/nhess-17-1683-2017, 2017.

———————————————————

---

## Author Comment (AC1) · 20 Mar 2018

Authors' responses to reviewer #1

V. Röthlisberger et al.  veronika.roethlisberger@giub.unibe.ch We would like to thank reviewer 1 for the constructive feedback to our manuscript. We much appreciate all comments and suggestions and will adopt most of them without reservation. Please find below all reviewer's comments and the authors' replies. RC1_1: In the introduction, a more general discussion about valuation methods and their application fields is missing. Commonly, approaches based on replacement values are distinguished from

approaches that rely on depreciated values. Insurance values or market values can be used to approximate one or the other. A distinction of these approaches and their application fields (e.g. insurance claims, cost-benefit-analysis) should be added. This issue should also be reflected later in the discussion. In the current paper, this issue is only very briefly mentioned in section 2.4.1, which is late and not sufficient in depth.

ARto_ RC1_1: Based on your comment, we will explicitly mention our valuation method (replacement values) in the introduction and briefly reflect the transferability of our results to other valuation methods (depreciated values) in the discussion.

RC1_2: The five models are well explained, but the rationales/justifications behind these models remain unclear. Please add some more background and assumptions about all models. Tab. 1 provides a comprehensive overview, but needs in my view more explanation in the main text. The same holds for Fig. 1.

ARto_ RC1_2: We will strengthen the rationales of all models in section 2.1 (Models' set-up for value estimation) and add more references to Tab. 1. More explanation on Fig. 1 will be given in section 2.4 (Data).

RC1_3: The authors find big differences between the Swiss unit costs and other published unit costs and explain this by differences in building standards and higher construction costs in Switzerland. It would be helpful to add some additional (real) data or statistics that underpin this explanation.

ARto_ RC1_3: We will add information and the following reference: Diaz Muriel, C.: Wide spread in construction prices across Europe in 2007, Eurostat, statistics in focus, 114/2008, 2008.

RC1_4: An overview table with advantages and disadvantages of all five models and their suitable applications would be helpful to summarize the findings (as a kind of counterpart to Table 1).

ARto_ RC1_4: We will take up this idea and we will add a table in section 3.4 (overall

discussion of the five models) that summarizes the core features (advantages / disadvantages) and suitable application of the five models.

RC1_5: The conclusion should end with an outlook on future research perspectives. (The implications of the results are addressed sufficiently.)

ARto_ RC1_5: We will provide an outlook on further research perspective at the very end of section 4 (conclusion).

Minor comments

RC1_Min1: In the introduction, a few important papers in this field are missing from my point of view, e.g.: - Barredo, J. I. (2009): Normalised flood losses in Europe: 1970–2006. - Nat. Hazards Earth Syst. Sci. 9: 97-104. - Jongman, B., Koks, E. E., Husby, T. G., and Ward, P. J. (2014): Increasing flood exposure in the Netherlands: implications for risk financing. - Nat. Hazards Earth Syst. Sci. 14: 1245-1255. - Kleist, L., A.H. Thieken, P. Köhler, M. Müller, I. Seifert, D. Borst, U. Werner (2006): Estimation of the regional stock of residential buildings as a basis for comparative risk assessment for Germany. – Nat. Hazards Earth Syst. Sci. 6: 541-552. - Seifert, I., A.H. Thieken, M. Merz, D. Borst, U. Werner (2010): Estimation of industrial and commercial assets values for hazard risk assessment. – Natural Hazards 52: 453- 479.

ARto_ RC1_Min1: We will check and possible add Barredo (2009), Kleist et al. (2006) and Seifert et al.; Jongman et al. (2014) are already addressed in the introduction.

RC1_Min2: Instead of "annual expected loss" either "expected annual damage" (EAD) or "average annual loss" (AAL) should be used.

ARto_ RC1_Min2: We will use the term "expected annual damage".

RC1_Min3: All abbreviations should be explained in the text once (e.g. AIC).

ARto_ RC1_Min3: We will check this point in the entire manuscript.

RC1_Min4: The hazard levels (high, medium, low) should be explained for readers

Interactive
comment

who are not familiar with the Swiss hazard zones.

ARto_ RC2_Min4: We will explain that briefly.

RC1_Min5: The correctness of the terms "global sums" or "global values" in section 3.4 should be checked. These sound a bit weird in this context.

ARto_ RC1_Min5: We will revise the manuscript accordingly.
* * *

---

## Author Response (AR1)

**Authors's reply to the reviewers's comments**

**Authors' point to point reply to all comments**

We would like to thank the two reviewer 2 for the constructive feedback to our manuscript. We much appreciate all comments and suggestions and adopted most of them without reservation. Please find below all reviewer's comments and the authors' replies and changes. References are made to the cleaned version of the revised manuscript. A manuscript version with tracked changes is added after the authors's point to point reply in this file.

RC1_1: In the introduction, a more general discussion about valuation methods and their application fields is missing. Commonly, approaches based on replacement values are distinguished from approaches that rely on depreciated values. Insurance values or market values can be used to approximate one or the other. A distinction of these approaches and their application fields (e.g. insurance claims, cost-benefit-analysis) should be added. This issue should also be reflected later in the discussion. In the current paper, this issue is only very briefly mentioned in section 2.4.1, which is late and not sufficient in depth.

ARto_ RC1_1: Based on your comment, we explicitly mentioned our valuation method (replacement values) in the introduction (p 3, l 25ff) and briefly reflected the transferability of our results to other valuation methods (depreciated values) in the conclusions (p 16, l 7ff).

RC1_2: The five models are well explained, but the rationales/justifications behind these models remain unclear. Please add some more background and assumptions about all models. Tab. 1 provides a comprehensive overview, but needs in my view more explanation in the main text. The same holds for Fig. 1.

ARto_ RC1_2: We strengthened the rationales of all models in section 2.1 (Models' set-up for value estimation, p 4, l 13ff ) and added more references to Tab. 1. More explanation on and reference to Fig. 1 is now given in section 2.4 (Data).

RC1_3: The authors find big differences between the Swiss unit costs and other published unit costs and explain this by differences in building standards and higher construction costs in Switzerland. It would be helpful to add some additional (real) data or statistics that underpin this explanation.

ARto_ RC1_3: We added information (p 10, l 14ff) and the following reference: Diaz Muriel, C.: Wide spread in construction prices across Europe in 2007, Eurostat, statistics in focus, 114/2008, 2008.

RC1_4: An overview table with advantages and disadvantages of all five models and theirsuitable applications would be helpful to summarize the findings (as a kind of counterpartto Table 1).

ARto_ RC1_4: We added a table (Tab 8) in section 3.4 (overall discussion of the five models) that summarizes the core features (advantages / disadvantages) and suitable application of the five models.

RC1_5: The conclusion should end with an outlook on future research perspectives. (The implications of the results are addressed sufficiently.)

ARto_ RC1_5: We added an outlook on further research perspective at the very end of section 4 (conclusion, p 17, l 3ff).

Minor comments

RC1_Min1: In the introduction, a few important papers in this field are missing from my point of view, e.g.:
- Barredo, J. I. (2009): Normalised flood losses in Europe: 1970–2006. - Nat. Hazards
Earth Syst. Sci. 9: 97-104.

- Jongman, B., Koks, E. E., Husby, T. G., and Ward, P. J. (2014): Increasing flood exposure in the Netherlands: implications for risk financing. - Nat. Hazards Earth Syst.
Sci. 14: 1245-1255.
- Kleist, L., A.H. Thieken, P. Köhler, M. Müller, I. Seifert, D. Borst, U. Werner (2006):
Estimation of the regional stock of residential buildings as a basis for comparative risk assessment for Germany. – Nat. Hazards Earth Syst. Sci. 6: 541-552.
- Seifert, I., A.H. Thieken, M. Merz, D. Borst, U. Werner (2010): Estimation of industrial and commercial assets values for hazard risk assessment. – Natural Hazards 52: 453-479.

ARto_ RC1_Min1: We added  Barredo (2009) and Kleist et al. (2006); Jongman et al. (2014) have already been addressed in the introduction.

RC1_Min2: Instead of "annual expected loss" either "expected annual damage" (EAD) or "average annual loss" (AAL) should be used.
ARto_ RC1_Min2: We now use the term "expected annual damage" (p2, l 13).

RC1_Min3: All abbreviations should be explained in the text once (e.g. AIC).
ARto_ RC1_Min3: We checked this point in the entire manuscript and revised accordingly (e.g. p 5, l 26). In addition we added Table A3 with all abbreviations used in the text.

RC1_Min4: The hazard levels (high, medium, low) should be explained for readers who are not familiar with the Swiss hazard zones.
ARto_ RC2_Min4: We revised the text accordingly (p 8, l 18ff).

RC1_Min5: The correctness of the terms "global sums" or "global values" in section 3.4 should be checked. These sound a bit weird in this context.
ARto_ RC1_Min5: We revised  the manuscript (p 15, l 17ff).

RC2_1: The title is interesting, I like it. However I feel like the connection between the content of the paper and the title isn't very good yet. The role of building value models in flood risk analysis gets very little attention compared to building value models in general. Could you either come up with a new title or reframe the abstract, introduction and conclusion a bit so the link between the paper and the title is better.
RC2_6: In this paper a clear separation is made between exposure and vulnerability. This is common in the literature. However, exposure values can also be combined with the vulnerability in flood risk analysis, for example in an absolute damage functions. This has the advantage that you only need one model rather than an exposure value model and then a model to estimate the damage fraction to be multiplied by this exposure value. An example of this setup is Wagenaar et al. (2017) which used Machine Learning methods and damage data to directly estimate the damage. These estimates are based on both hazard and exposure characteristics. Implicitly such models therefore also include an exposure valuation model. Could you discuss the benefits of the two step approach taken in this study (exposure value separated from vulnerability)?

ARto_ RC2_1 and ARto_ RC2_6: We revised the manuscript, covering the highlighted issues as follows: In section 1 (introduction), more attention is paid to the role of building value models in flood risk analyses and on alternative approaches (models using absolute damage functions) (p2, l 13ff) . Section 4 (conclusions, p 16, l 5 ff) and the abstract (p 1, l 31ff) are revised accordingly.

RC2_2: On Page 9, line 20 you state that attached buildings are counted as one building, this is quite different from other models that I know. Maybe this is why you find the extreme values so important? I can imagine uniform models perform really bad because of this. This also makes the building stock of a particular area very important for the model performance with a

uniform value. Maybe discuss this assumption behind the paper a bit better because it could make the results not applicable to other areas.

ARto_ RC2_2: Our study clearly shows that there is a high correlation between the volume and the reconstruction costs of a building. Provided that there is a spread in the volume of a particular building stock, models that consider the building volume (M2, M4, M5) outperform models that do not consider the building volume (M3 and M5). The reviewer is right, that the difference between the two model groups increases with the spread of the building volumes of a particular building stock. However, as long as there is any spread in the building volumes the models that do not consider the building volume underestimate the values in flood areas with comparable high buildings volumes as well as overestimate the values in flood areas with comparable low building volumes. Thus, our results on differences between the models are valid for any area with buildings of different volumes. We highlighted this aspect in more detail in the new version of the manuscript (Section 3.4, p 15, l 12ff; Section 4, p 16, l 20ff).

RC2_3: If I understand it correctly you first determined which model performs best for the places that have good data available. Then you compare the other models to this best "benchmark" model for the whole country. Why do you take this two-step approach? Why not just compare everything to the available insurance data points and make your judgment on this?

ARto_ RC2_3: One purpose of our study is the comparison of exposure figures at the national scale, including their spatial and statistical distribution. For this reason, we not only compare the results in (the fragmented area of) the eight Cantons with comprehensive insurance data sets but for entire Switzerland. However, we agree with the reviewer that the main statements / conclusions are the same for the eight Cantons as for entire Switzerland. We did not change the manuscript based on this comment.

RC2_4: How accurate is the insurance data that the benchmark model was picked on? Are these values also based on a model or are these expert estimates? If expert estimates, do these experts have some valuation model that they apply? My worry is that the insurance data has artificial relationships in it (based on their valuation model) and that this study is just recreating the valuation model of the insurance companies. In that case the model that is closest to the currently applied insurance models performs best. Can you please discuss to what extend this is a possibility?

ARto_ RC2_4: As much as we are informed by our data providers (i.e., the insurance companies), insurance values are object-specific estimates by experts that are (a, for new buildings) based on documented construction costs (invoices) or (b, for older buildings) based on on-site inspection and validation. In general, these insurance values are highly confident and not publicly available. To fill the gap regarding building values in exposure (and risk) models, we create and compare models, which use publicly available data. It is possible, that insurance companies implicitly use comparable models based on similar parameters. However, even if such models were used by some companies, they are not published and hence a comparison with our approaches is not possible. We added the method of value estimation to the manuscript (section 2.4.1, p 7, l 29ff)

RC2_5: The conclusion states that M1 and M3 underestimate the exposure values. This sounds like a bias in the model. Is this bias not simply a problem with the parameter values rather than a problem with the model itself? If you would just increase the parameters values wouldn't that get rid of the entire bias? Or I think I probably misunderstood what you meant by this sentence so please try to explain this a bit better.

ARto_ RC2_5: The found underestimation of exposure values by M1 and M3 means that buildings exposed to flood are in general bigger than the overall building stock. Thus, as the found correlation between volume and replacement costs suggests, these larger buildings in the flood plains have higher replacement values and are underestimated by M1 and M3 which do not consider the building size. We added this comment to section 4 (conclusion, p 16, l 20ff).

Minor comments

50

RC2_Min1: Page 1, line 24. Please explain how risk management focuses on extreme exposure values. I'm a bit skeptical about this so please convince me.

ARto_ RC2_Min1: In Switzerland, as well as in other countries, decisions of public investments into flood reduction measures are based on quantitative cost-benefit analyses. This implies that areas (i.e. floodplains) with a high flood risk (high hazard probability, high exposure, and high vulnerability) are prioritized over others. As exposure is one important factor of risk analysis, our study contributes to priority setting in flood risk management, although further intersections with hazard and vulnerability are required in a further step. We did not change the manuscript based on this comment.

RC2_Min2: Page 5, line 28. Please explain what a hexagon is in this context.

ARto_ RC2_Min2: A hexagon is a 2D geometrical feature with six edges and six sides of equal length. Here, we used hexagons to divide the total area into smaller entities with equal size and shape. With this procedure, we addressed the modifiable area unit problem MAUP (see Röthlisberger et al. 2017).

Röthlisberger, V., Zischg, A.P., Keiler, M., 2017. Identifying spatial clusters of flood exposure to support decision making in risk management. Science of The Total Environment 598, 593–603.
We did not change the manuscript based on this comment.

RC2_Min3: Page 12, line 19. Please explain homoscedastic.

ARto_ RC2_Min3: In this context, homoscedastic means that the factors (of differences between exposure value based on model M2 (or M4 or M5) and the direct application of insurance values) are not dependent on the exposure value, the "variance of the factors" are the same for hexagons with low and high exposure values.
We did not change the manuscript based on this comment.

RC2_Min4: Figure 1 and the data selection section (2.4.2) are currently difficult to understand.
Please start with why data selection is required and then lead the reader along figure 1 explaining at every step why each action is carried out.

ARto_ RC2_Min4: We revised section 2.4.2, see also our comment "ARto_ RC1_2" in the response to reviewer #1.

RC2_Min5: Establish a clear definition of building value in the introduction. You mention it is replacement value later on but maybe move that to the introduction. Also a good definition of replacement value is important and useful (can also be a reference). This seems quite relevant for this paper.

ARto_ RC2_Min5: We revised the introduction section mentioning the value type already at this point (p 3, l 25 ff).

RC2_Min6: In section 2.2 please explain that in your definition of exposure only buildings are included that can actually flood. I also know definitions of exposure in which any building is included.

ARto_ RC2_Min6: We revised section 2.2. accordingly (p 6, l 16ff).

RC2_Min7: In 2.4.1 a whole list of abbreviations is introduced at once. This is difficult to follow
and makes the text a bit of a puzzle (especially on the first read). So I would choose not to use abbreviations in this case.

ARto_ RC2_Min7: We keep the abbreviations because they link the text with the figures (especially Fig.1) but we added a table (Table A3) with all abbreviations used in the text.

5   RC2_Min8:   Page 6, line 12-18. Please explain better why this approach is required. I don't get why you need the entire benchmarking next to the validation on insurance data.

ARto_ RC2_Min8:  please refer to our comment "ARto_ RC2_3" above.

RC2_Min9:  Page 12, line 30. MEA is I think MAE

ARto_ RC2_Min9: We revised accordingly (p 13, l25)

[revised manuscript text omitted]

---

## Referee Report (RR1)

**Review "On the role of building value models for flood risk analysis"**

The changes to the manuscript are minimal and I'm not fully satisfied with what has been done with the comments. The current manuscript however doesn't contain important flaws. The research has been carried out well and is new but at the same time a bit obvious and all the conclusions are as expected. Therefore, I have a couple of points to make the manuscript both better and more interesting.

**Important points:**

- The link between the title and the content is still a bit poor. The reviewer added to the abstract and conclusions that there is a focus on exposure. However, the title suggests that the paper is some sort of sensitivity analysis to see how important building values are in flood risk analysis. Some of that is done in the paper but within the current framing of the paper this comes through very little. It would be good if you could let this aspect of the research come back a little bit more. Because given the current content I think a better title would be: "A comparison of building value models for flood risk analysis".
- The point that the conclusions are only valid if there is a significant spread in building volumes is maybe good to repeat in the conclusion. This is a really major point because the assumption that attached buildings are one building increases this spread a lot and hence makes the difference between the models in the conclusions much larger. This is not a very common assumption also so it might make the conclusions not applicable elsewhere.

**Suggestion:**

- I like some discussion on the future of building value models for flood risk assessment. In flood vulnerability modelling there is a trend to use Machine Learning methods to assess the vulnerability (e.g. Merz et al., 2012). In Wagenaar et al. (2017) absolute damages are determined and hence the building values are already included in that model also in a multi-variable way. So many variables can contribute to the building value, such as building age for example. Do you envision in the future also more complex multi-variable models for building value alone? The M5 model is a first step in this direction, is it useful to get more detailed and are there important influencing variables that are not taken into account in these models (e.g. building age, building material, quality of maintenance) or maybe proxies such as income levels in of the inhabitants or area.

**References**

- Merz, B., Kreibich, H., and Lall, U.: Multi-variate flood damage assessment: a tree-based data-mining approach, Nat. Hazards Earth Syst. Sci., 13, 53-64, https://doi.org/10.5194/nhess-13-53-2013, 2013.
- Wagenaar, D., de Jong, J., and Bouwer, L. M.: Multi-variable flood damage modelling with limited data using supervised learning approaches, Nat. Hazards Earth Syst. Sci., 17, 1683-1696, https://doi.org/10.5194/nhess-17-1683-2017, 2017.

---

## Author Response (AR2)

**Authors' reply and manuscript with tracked changes**

We thank both reviewers for their constructive and positive feedback. Please find below all reviewer's comments and the authors' replies and changes. References are made to the cleaned version of the revised manuscript. A manuscript version with tracked changes is added after the authors's point to point reply in this file.

**Reviewer 1 (technical corrections)**

Dear authors,
thank you very much for considering my remarks and for revising the paper carefully. In my view, all my comments have been addressed, although I am surprised that cost-benefit-analyses in Switzerland are based on replacement costs. From an economic point of view depreciated values would be more suitable in such a context. You might want to comment on this, but since this is not the focus of the paper, a further revision is not needed.
The Tables 3 and 8 are very helpful. Thank you!
I was wondering whether your first affiliation (1: Institute of Geography and Mobiliar Lab for Natural Risks) is correct since the Mobiliar Lab for Natural Risks is again mentioned in affiliation 2.
Overall, I recommend accepting the paper in its current version and I am looking forward to seeing the paper published.

*Authors' response: We corrected the affiliation*

**Reviewer 2 (Minor revisions)**

**Important points:**
☐The link between the title and the content is still a bit poor. The reviewer added to the abstract and conclusions that there is a focus on exposure. However, the title suggests that the paper is some sort of sensitivity analysis to see how important building values are in flood risk analysis. Some of that is done in the paper but within the current framing of the paper this comes through very little. It would be good if you could let this aspect of the research come back a little bit more. Because given the current content I think a better title would be: "A comparison of building value models for flood risk analysis".

*Authors' response: We changed the title accordingly in the manuscript (p1, l1) and abstract*

☐The point that the conclusions are only valid if there is a significant spread in building volumes is maybe good to repeat in the conclusion. This is a really major point because the assumption that attached buildings are one building increases this spread a lot and hence makes the difference between the models in the conclusions much larger. This is not a very common assumption also so it might make the conclusions not applicable elsewhere.

*Authors' response: We repeated the remark in the conclusion (Sec 4, p16, l25)*

**Suggestion:**

☐I like some discussion on the future of building value models for flood risk assessment. In flood vulnerability modelling there is a trend to use Machine Learning methods to assess the vulnerability (e.g. Merz et al., 2012). In Wagenaar et al. (2017) absolute damages are determined and hence the building values are already included in that model also in a multi-variable way. So many variables can contribute to the building value, such as building age for example. Do you envision in the future also more complex multi-variable models for building value alone? The M5 model is a first step in this direction, is it useful to get more detailed and are there important influencing variables that are not taken into account in these models (e.g. building age, building material, quality of maintenance) or maybe proxies such as income levels in of the inhabitants or area.

*Authors' response: We extended the conclusions accordingly (Sec 4, p17, l5ff) and referenced to Wagenaar et al. 2017*

[revised manuscript text omitted]